# Different adaptive NO-dependent Mechanisms in Normal and Hypertensive Conditions

**DOI:** 10.3390/molecules24091682

**Published:** 2019-04-30

**Authors:** Michaela Kosutova, Olga Pechanova, Andrej Barta, Sona Franova, Martina Cebova

**Affiliations:** 1Institute of Normal and Pathological Physiology, Centre of Experimental Medicine SAS, Sienkiewiczova 1, 813 71 Bratislava, Slovak Republic; michaela.kosutova@savba.sk (M.K.); olga.pechanova@savba.sk (O.P.); andrej.barta@savba.sk (A.B.); 2Department of Pharmacology, Biomedical Centre JFM CU, Comenius University Bratislava, Jessenius Faculty of Medicine Martin, Malá Hora, 11161/4C, 036 01 Martin, Slovak Republic; sona.franova@jfmed.uniba.sk

**Keywords:** myocardial infarction, L-NAME, nitric oxide

## Abstract

Myocardial infarction (MI) remains the leading cause of death worldwide. We aimed to investigate the effect of NO deficiency on selective biochemical parameters within discreet myocardial zones after experimentally induced MI. To induce MI, the left descending coronary artery was ligated in two groups of 16-week-old WKY rats. In one group, NO production was inhibited by L-NAME (20 mg/kg/day) administration four weeks prior to ligation. Sham operations were performed on both groups as a control. Seven days after MI, we evaluated levels of nitric oxide synthase (NOS) activity, eNOS, iNOS, NFҡB/p65 and Nrf2 in ischemic, injured and non-ischemic zones of the heart. Levels of circulating TNF-α and IL-6 were evaluated in the plasma. MI led to increased NOS activity in all investigated zones of myocardium as well as circulating levels of TNF-α and IL-6. L-NAME treatment decreased NOS activity in the heart of sham operated animals. eNOS expression was increased in the injured zone and this could be a compensatory mechanism that improves the perfusion of the myocardium and cardiac dysfunction. Conversely, iNOS expression increased in the infarcted zone and may contribute to the inflammatory process and irreversible necrotic changes.

## 1. Introduction

Myocardial infarction (MI) remains the leading cause of death worldwide [1,2]. Pathologically, MI is defined as a prolonged ischemia with subsequent loss of cardiomyocytes in the infarcted area. Left ventricular (LV) remodeling after MI is characterized by necrosis, excessive fibrosis, cardiomyocyte hypertrophy, inflammation and loss of capillaries, which can eventually lead to heart failure [3,4]. Restoration of the blood supply to the ischemic myocardium is essential for limiting the damage caused by acute myocardial infarction. Paradoxically, the reperfusion of ischemic tissue may result in additional cardiomyocyte dysfunction as a consequence of cellular enzymes released during re-oxygenation, this phenomenon is termed ischemia-reperfusion (I/R) injury [5]. The mechanism by which reperfusion contributes to additional cardiac damage is still under intense discussion. There are three major contributors to I/R injury: 1) reactive oxygen species (ROS), 2) deranged Ca^2+^ signaling and 3) destruction of mitochondrial integrity and function. The relative imbalance between the generation and the decomposition of ROS in ischemic tissue induces pro-inflammatory cascades by activating multiple protein complexes in myocardial cells [6]. In the myocardium, nuclear factor-erythroid 2 (Nrf2) is a critical element of redox homeostasis [7]. In addition to protection against oxidative stresses, Nrf2 responds to pro-inflammatory stimuli and protects against inflammatory injuries and fibrosis [8,9]. Elevated Ca^2+^ levels can trigger hypercontracture and the activation of deleterious signaling cascades [10]. The contributor to I/R injury is the mitochondrial membrane, which in addition to producing energy can also signal cell death and cardiomyocyte disintegration [11].

Nitric oxide (NO) has been a hotspot in cardiovascular research for decades. NO is endogenously produced within myocardium via either NOS-dependent or NOS-independent mechanisms. Four isoforms of NOS have been described: neuronal NOS (nNOS), mainly located in cardiac myocytes; endothelial NOS (eNOS), typically expressed in coronary and cardiac endothelium; inducible NOS (iNOS), originating from neutrophils or myocytes during inflammation; and mitochondrial NOS (mtNOS), present in cardiac mitochondria [12]. NOS activity is regulated in several ways including compartmentalization, co-factor and substrate availability, transcriptional and translational modulations [13]. It is generally accepted, that NO is an essential signaling molecule involved in many physiological processes in animals and humans, including neurotransmission, hypertension and heart failure. It is also a known trigger and mediator for cardioprotection [14], reducing myocardial necrosis and apoptosis [15,16]. On the other hand, in pathophysiological conditions such ischemia, the accumulation of NO from both enzymic and non-enzymic sources may play a significant role in I/R injury by increasing ROS production [17]. Based on this, there is a potential role of endogenously activated Nrf2 and eNOS in MI-induced oxidative stress [18,19].

Chronic inhibition of NOS by NG-nitro-L-arginine methyl ester (L-NAME) is a well-established model of experimental hypertension and organ damage characterized by myocardial hypertrophy, fibrosis and coronary artery wall hyperplasia [20,21,22,23]. Myocardial infarction prognosis is conditioned by a series of risk factors which renders the process more incisive. If hypertension is present at time of myocardial infarction, it worsens the prognosis of MI by inducing a burden of oxidative and inflammatory mediators released within the heart. Therefore, studies with molecules possessing multiple activities and thus capable of blocking or reversing the pathological progress of hypertension and MI at different levels are needed. Nitric oxide represents an important part of this strategy. Generally, NO is an essential signaling molecule involved in many physiological and pathophysiological processes. Therefore, it is of interest to use a model of L-NAME induced hypertension for experimental myocardial infarction. On the one hand, the unfavorable conditions of hypertension due to four weeks of NO synthase inhibition may significantly worsen the consequences of myocardial infarction. On the other hand, lack of nitric oxide could paradoxically have a positive effect on heart ischemia. Therefore, the aim of the study was to block NO production with L-NAME administration four weeks prior to experimentally induced myocardial infarction by coronary artery ligation and thereafter to evaluate the effect of NO deficiency on selective biochemical parameters within individual myocardial zones. 

## 2. Results

### 2.1. Blood Pressure, Body Weight, Cardiac and Kidney Weight, Cytokine Levels

Blood pressure increased continually after L-NAME administration from the first until the last week of the experiment. Significantly increased blood pressure (*p* < 0.01) in both L-NAME groups persisted after surgery. There were no differences in blood pressure within the groups in normotensive rats (see Table 1).

Body weight did not differ between L-NAME + Sham rats and normotensive WKY + Sham controls during the experiment or after coronary ligation. The total weight of the heart and the left kidney further indicated no differences within these groups (see Table 2).

TNF-α and IL-6 were increased by 125% and 79% respectively, after myocardial infarction in the normotensive group (WKY + MI) vs. WKY + sham group (*p* < 0.01). There was also an increase in TNF-α and IL-6, 130% and 46% respectively (*p* < 0.01, Table 3) in the L-NAME + MI animals when compared to L-NAME + sham rats. 

### 2.2. Total NOS Activity 

Total NO-synthase activity was not changed within all three investigated myocardial zones, infarcted zone (IZ), injured zone (INZ) and non-infarcted zone (NIZ), in normotensive rats after experimentally induced MI (WKY + MI) when compared to the normotensive sham group. A four-week-long L-NAME treatment (L-NAME + Sham) significantly decreased NOS activity compared to WKY + Sham (*p* < 0.05). NOS activity level in L-NAME group after myocardial infarction (L-NAME + MI) remained decreased in NIZ, but it was increased in IZ and INZ in comparison to L-NAME + sham group (*p* < 0.05; Figure 1A–C). 

### 2.3. Protein Expression Levels

Western blot analysis was used to determine protein expression levels within each of the four groups. Endothelial NOS protein expression was upregulated in IZ and INZ of WKY + MI group (*p* < 0.05; Figure 2A) as well as in INZ of L-NAME + MI group (*p* < 0.05; Figure 2B). The L-NAME only administration did not change eNOS expression in any zone of myocardium in comparison to the controls (Figure 2A–C). iNOS was downregulated in the infarcted zone in WKY + MI group (*p* < 0.05; Figure 3A) and the injured zone in L-NAME + MI group (Figure 3B). There were no differences in iNOS expression in NIZ of normotensive and hypertensive rats (Figure 3C).

Interestingly, Nrf2 protein expression levels within the INZ were significantly decreased (*p* < 0.05) in WKY + IM when compared to WKY + Sham animals (Figure 4B). Other values did not differ (Figure 4A,B,C). The present study further assessed the effect of inflammation after MI by determining NFҡB/p65 protein expression levels. NFҡB/p65 is upregulated in the IZ of both hypertensive groups, and significantly higher in L-NAME + MI group (*p* < 0.01; Figure 5A). Surprisingly, we observed a significant increase in NFҡB/p65 within the INZ and NIZ of L-NAME + sham rats (*p* < 0.01), however, it was downregulated in the INZ in L-NAME + MI animals (*p* < 0.01, Figure 5B,C).

## 3. Discussion

Myocardial infarction is a serious coronary heart disease associated with ventricular remodeling as a consequence of myocardial ischemia. Myocardial ischemia refers to the pathological state of reduced oxygen supply and the accumulation of residual metabolites as a result of decreased perfusion. The absence of oxygen and nutrients during ischemia creates conditions resulting in inflammation and oxidative damage.

Nitric oxide is well-established as a trigger and mediator of cardioprotection and chronic inhibition of NO, by L-NAME, induces hypertension. Our results show continually increased blood pressure following L-NAME administration, persisting after coronary occlusion. This is consistent with our previous study and with others [15,24,25]. Experimentally induced myocardial infarction in normotensive animals did not affect NOS activity in any investigated myocardial zones. Administration of a nonselective NOS inhibitor, L-NAME, decreased NOS activity in all three myocardial zones; however, coronary occlusion resulted in increased level of NOS activity in IZ and INZ similar to the increases seen in normotensive rats. The present results suggest that following MI, increased NOS activity in IZ and INZ could act as a compensatory mechanism improving the perfusion of the myocardium. It could prevent further cardiac dysfunction and development of heart failure. 

Upregulation of NOS may be associated with activation of individual isoforms. In general, eNOS and nNOS are constitutively expressed, whereas iNOS is predominantly expressed after inflammation stimulus. In the current study, eNOS expression in IZ and INZ was significantly increased after MI and this is consistent with previous studies [26] which demonstrated increased eNOS expression in the coronary arteries of infarcted rats. NO generated by eNOS is important in regulation of blood pressure and it may globally influence cardiac function and remodeling. iNOS was only increased in the IZ in L-NAME + MI rats, but it was decreased in the INZ of the same group. Similar results were published by Chen et al. [27] who found increased iNOS expression in cardiac cytosolic samples in rats with MI induced by isoproterenol treatment. Our results indicate that eNOS upregulation in INZ after MI in L-NAME rats may contribute to increased NOS activity and serve as a compensatory mechanism improving perfusion of the myocardium. Conversely, within the INZ, iNOS expression was increased in L-NAME + MI rats and this isoform contributed to increased NOS activity in IZ after MI in L-NAME rats. Moreover, NF-kappaB expression paralleled this pattern of iNOS expression. Thus, increased NOS activity in this case may be further elevated to the pathophysiological level and may enhance inflammatory processes with irreversible necrotic changes. According to Chen and co-authors such processes may significantly contribute to the formation of myocardial scar after myocardial infarction [27]. Nuclear factor kappa B (NFҡB) signaling has been implicated as a possible mechanism responsible for upregulation of both eNOS and iNOS [28,29]. In our study, decreasing NO by L-NAME treatment increased NFҡB expression. Increased NFҡB levels can lead to structural remodeling, changing the physiological properties of the heart, contributing to cardiac dysfunction. NFҡB can also be activated by up-regulation of TNF-α [30]. Elevated serum and cardiac TNF-α concentrations correlate to the depression of myocardial function in ischemia/reperfusion [31]. Inflammation is the main pathological process in the early period of myocardial infarction. In the present study, myocardial infarction increased the TNF-α and IL-6 levels in normotensive and L-NAME-induced hypertensive rats. Similar results were found in human studies where the cytokines TNF-α, IL-1β, and IL-6 levels were significantly higher in the patients after MI compared to control [32]. Wang and colleagues [33] showed that the highest level of TNF-α, MCP-1, IL-6, and IL-1β mRNA levels and IL-6 and TNF-α protein levels appeared to be maximal at 1 week after coronary artery ligation in rats but began to subside thereafter. High level of TNF-α and IL-6 further potentiates procoagulant effects by increasing the reactivity of platelets [34]. Our results are consistent with this data, illustrating that ischemic damage of myocardial tissue can lead to production and release of proinflammatory cytokines.

The Nrf2 pathway is a major regulator in inhibiting the oxidative stress of cellular redox conditions. Under oxidative stress, activation of Nrf2 genes leads to an increase in expression and activity of antioxidant enzymes. Therefore, Nrf2 is an important pathway in diminishing ROS produced during IR. However, in this study we did not see a diminished Nrf2 protein level after myocardial infarction. Our study shows, that damaged biochemical markers after myocardial infarction are mediated via NFҡB signaling pathway rather than Nrf2. Our data also suggests that NFκB may negatively regulate Nrf2 expression by inactivating Nrf2 via competitive interaction with the CH1-KIX domain of CREB binding protein. These observations illustrate the complicated interplay between the NFκB and Nrf2 pathways [35].

## 4. Materials and Methods 

### 4.1. Animals and Treatment

The experiments were carried out on male 16-week-old Wistar Kyoto rats housed in a room with a maintained temperature (22 ± 2 °C), relative humidity (55 ± 10%), and 12-h light / dark cycle. The animals were obtained from Velaz laboratories (Prague, Czech Republic). All procedures and experimental protocols were performed in accordance with institutional guidelines and were approved by the State Veterinary and Food Administration of the Slovak Republic (Ro-3037/14-221) and by an Ethical committee of the Centre of Experimental Medicine SAS according to the European Convention for the Protection of Vertebrate Animals used for Experimental and other Scientific Purposes, Directive 2010/63/EU of the European Parliament.

Rats were divided into four groups: 1. control sham operated 16-week-old male WKY rats (WKY + sham); 2. age-matched WKY rats with experimentally induced myocardial infarction (WKY + MI); 3. sham operated 16-week-old male WKY rats with L-NAME induced hypertension (20 mg /kg/day of L-NAME from the 12th; L-NAME + sham); 4. 16-week-old male WKY rats with L-NAME induced hypertension and experimentally induced myocardial infarction (20 mg /kg/day of L-NAME from the 12th; L-NAME + MI) (n=6 in each group). L-NAME (Sigma-Aldrich) was dissolved in the drinking water at a dose of 20 mg / kg / day and administered daily to 12-week-old normotensive rats for 4 weeks. Normotensive control rats had free access to drinking water and all animals had free access to standard lab chow (Altromin 1324P). Blood pressure was measured non-invasively in pre-warmed animals by tail-cuff plethysmography (The IITC Life Science, MRBP blood pressure system) every week from the age of 10 weeks. The measurement was taken in triplicates. 

### 4.2. Experimentally Induced Myocardial Infarction

Experimental MI was induced in 16-week-old normotensive and L-NAME hypertensive rats by ligation of the left descending coronary artery. Prior to surgery, the analgesia was applied with active substance Butorphanol in the dose 2 mg/kg s.c. + 2 mg/kg meloxicam with 5 mL saline + 5% glucose s.c. After administration of anesthesia (i.p. with titelamine-zolazepan 30 mg/kg), the rats were intubated and ventiled with a pressure-controlled rodent respirator at 70 strokes per minute. Rats were placed on an electric heating pad to maintain a body temperature of 37 °C. The left lateral thoracotomy was performed through the fifth intercostal space. After opening the pericardium, the left coronary artery was reversible ligated with silk 5-0; C-2 suture thread (Ethicon, San Lorenzo, CA, USA) at approximately 2 mm from its origin. MI was confirmed by ECG. Representative ECG recording is shown on Figure 6A for a sham operated WKY rat. Equivalent recording was obtained with WKY rat after ligation to confirm MI (Figure 6B). Twenty minutes after inducing MI, the occlusion was released. Thereafter, thoracotomy was closed on layers using Vicryl 4-0; SH-1 and Prolene 8-0; CC (Ethicon, San Lorenzo, CA., USA). Sham operated animals underwent the same procedure without ligation. Seven days after inducing MI, animals were sacrificed by overdose of anesthesia (i.p. with titelamine-zolazepan 60mg/kg). BW (body weight), HW (heart weight) and kidney weight (KW) were determined. TNF-α and IL-6 levels were determined using commercial Bio-Plex Pro Cytokine kit in the plasma. 

### 4.3. Total NOS Activity 

Total NOS activity was determined in crude tissue homogenates from different parts of the heart – infarcted zone (IZ), injured zone (INZ) and non-infarcted zone (NIZ) by measuring [3H]-L- citrulline formation from [3H]-L- arginine (MP Biochemicals, St. Louis, MO, USA) as described elsewhere [36]. [3H]-L-citrulline content was measured by liquid scintillation calculation (TriCarb, Packard, UK). NOS activity is expressed as picokatal per gram of protein (pkat/g protein).

### 4.4. Western Blot Analysis

Tissue samples of each investigated myocardial zone were homogenized in lysis buffer –0.05 mM Tris containing protease inhibitor cocktail (Sigma-Aldrich, Germany). After centrifugation (15,000 rpm at 4 °C for 20 min) protein concentrations were determined by Lowry assay. Supernatants were subjected to SDS-PAGE using 12% gels to examine protein and transferred to nitrocellulose membranes. Membranes were blocked with 5% non-fat milk in Tris-buffer solution (TBS; pH 7.6) containing 0.1% Tween-20 (TBS-T) for 1 h at room temperature and probed with a primary polyclonal rabbit anti-endothelial NOS and anti-inducible NOS, anti-NF-kB, anti-Nrf2 antibodies and anti-GAPDH as a loading control (Abcam, Cambridge, UK) overnight at 4 °C. Antibodies were detected using a secondary peroxidase-conjugated anti-rabbit antibody (Abcam, Cambridge, UK) at the room temperature for 2 h. The intensity of bands was visualized using the enhanced chemiluminiscence system (ECL, Amersham, UK), quantified by using ChemiDocTM Touch Imagine System (Image LabTM Touch software BioRad, Hercules, CA, USA) and normalized to GAPDH bands. 

### 4.5. Statistical Analysis

All data are expressed as means ± S.E.M. One-way ANOVA and Duncan’s test was used for statistical analysis. *p* < 0.05 value was considered statistically significant.

## 5. Conclusion

In conclusion, our data shows that ligation of coronary arteries induced different effects in normal and hypertensive conditions. Our results suggest that increased eNOS expression/activity could be a compensatory mechanism that improves perfusion of the myocardium and cardiac dysfunction. On the other hand, increased iNOS expression/activity may lead to inflammatory process and irreversible necrotic changes. 

## Figures and Tables

**Figure 1 molecules-24-01682-f001:**
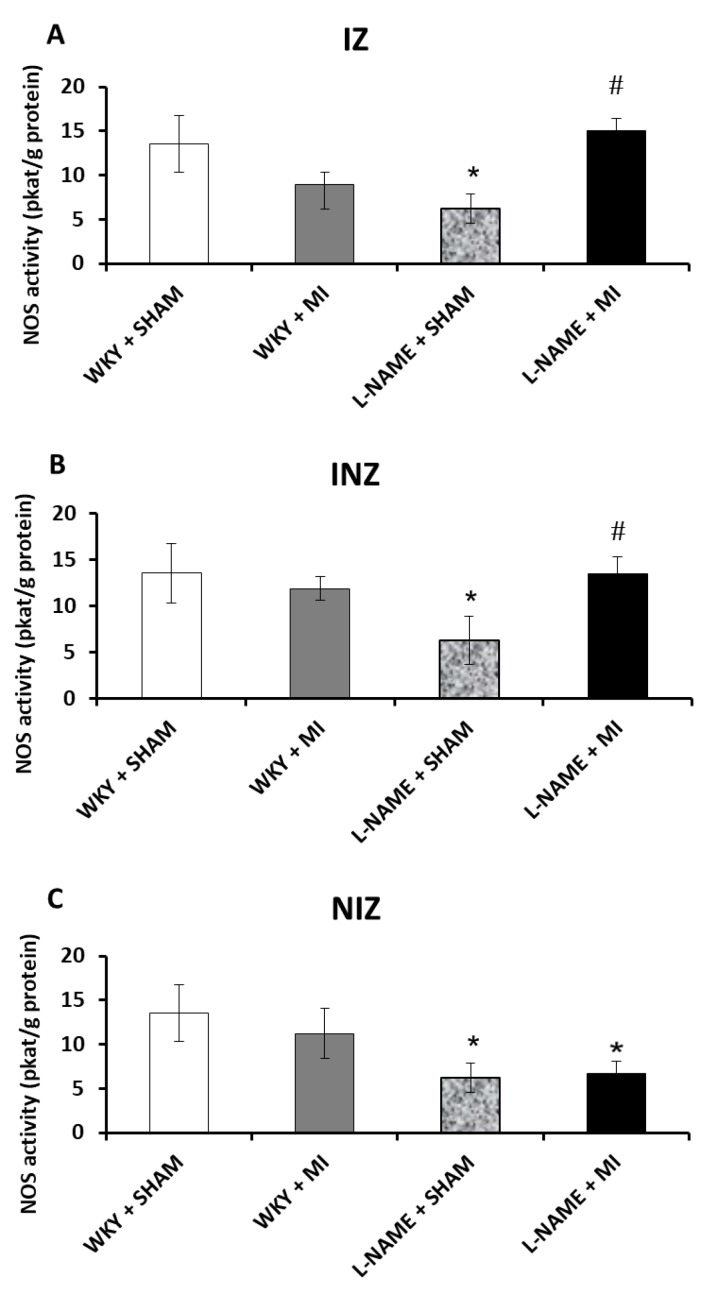
The effect of myocardial infarction on nitric oxide synthase (NOS) activity of infarcted zone (A), injured zone (B) and non-infarcted zone (C) of myocardium. Total NOS activity levels of sham operated Wistar Kyoto rats (WKY + Sham); WKY rats with experimentally induced myocardial infarction (WKY + MI); sham operated WKY rats treated with L-NAME (20 mg/kg/day) (L-NAME + Sham) and L-NAME (20 mg/kg/day) treated WKY rats with experimentally induced myocardial infarction (L-NAME + MI) in infarcted zone (IZ) (A), injured zone (INZ) (B) and non-infarcted zone (NIZ) (C). * *p* < 0.05 compared to WKY + Sham; # *p* < 0.05 compared to L-NAME + Sham group; N = 6 in each group.

**Figure 2 molecules-24-01682-f002:**
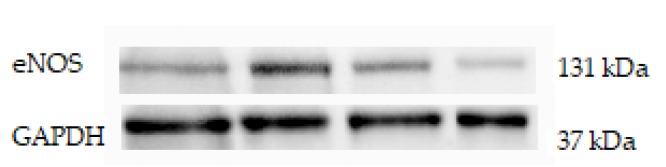
The effect of myocardial infarction on endothelial NOS (eNOS) expression of infarcted zone (A), injured zone (B) and non-infarcted zone (C) of myocardium. Representative western blot images and relative eNOS expression levels of sham operated Wistar Kyoto rats (WKY + Sham); WKY rats with experimentally induced myocardial infarction (WKY + MI); sham operated WKY rats treated with L-NAME (20 mg/kg/day) (L-NAME + Sham) and L-NAME (20 mg/kg/day) treated WKY rats with experimentally induced myocardial infarction (L-NAME + MI) in infarcted zone (IZ) (A), injured zone (INZ) (B) and non-infarcted zone (NIZ) (C). * *p* < 0.05 compared to WKY + Sham group; N = 6 in each group.

**Figure 3 molecules-24-01682-f003:**
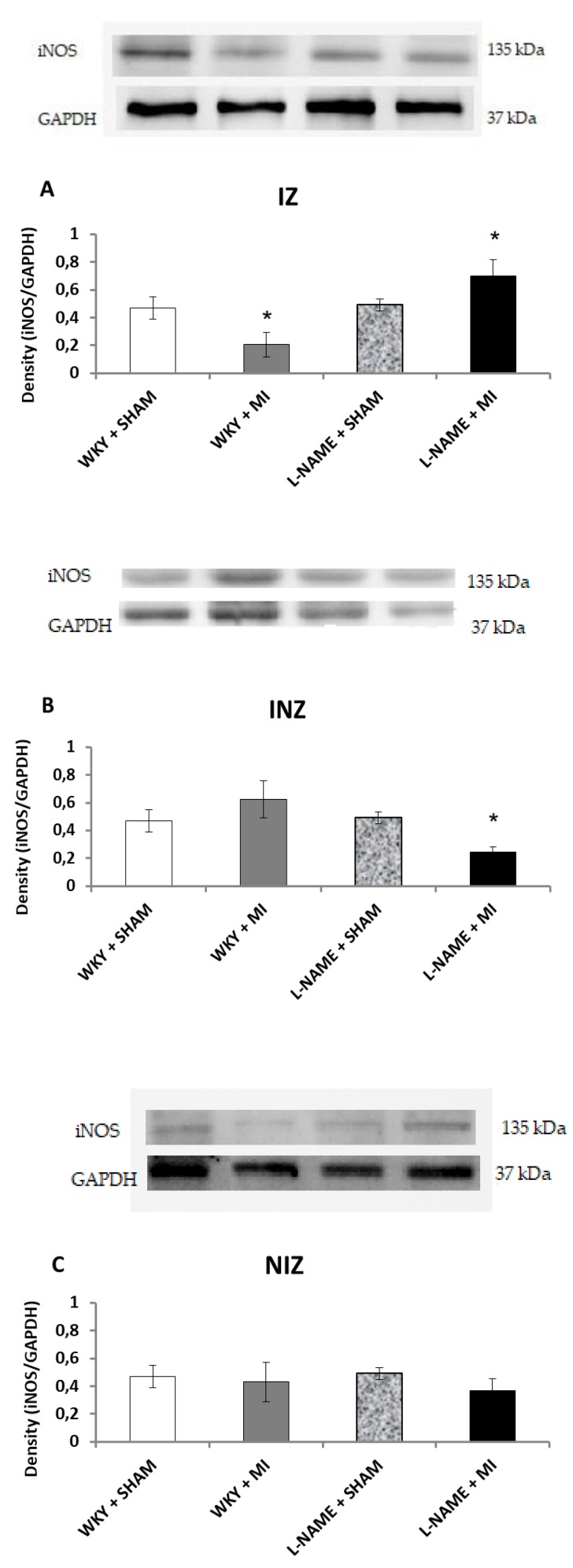
The effect of myocardial infarction on inducible NOS (iNOS) expression of infarcted zone (A), injured zone (B) and non-infarcted zone (C) of myocardium. Representative western blot images and relative iNOS expression levels of sham operated Wistar Kyoto rats (WKY + Sham); WKY rats with experimentally induced myocardial infarction (WKY + MI); sham operated WKY rats treated with L-NAME (20 mg/kg/day) (L-NAME + Sham) and L-NAME (20 mg/kg/day) treated WKY rats with experimentally induced myocardial infarction (L-NAME + MI) in infarcted zone (IZ) (A), injured zone (INZ) (B) and non-infarcted zone (NIZ) (C). * *p* < 0.05 compared to WKY + Sham group; N = 6 in each group.

**Figure 4 molecules-24-01682-f004:**
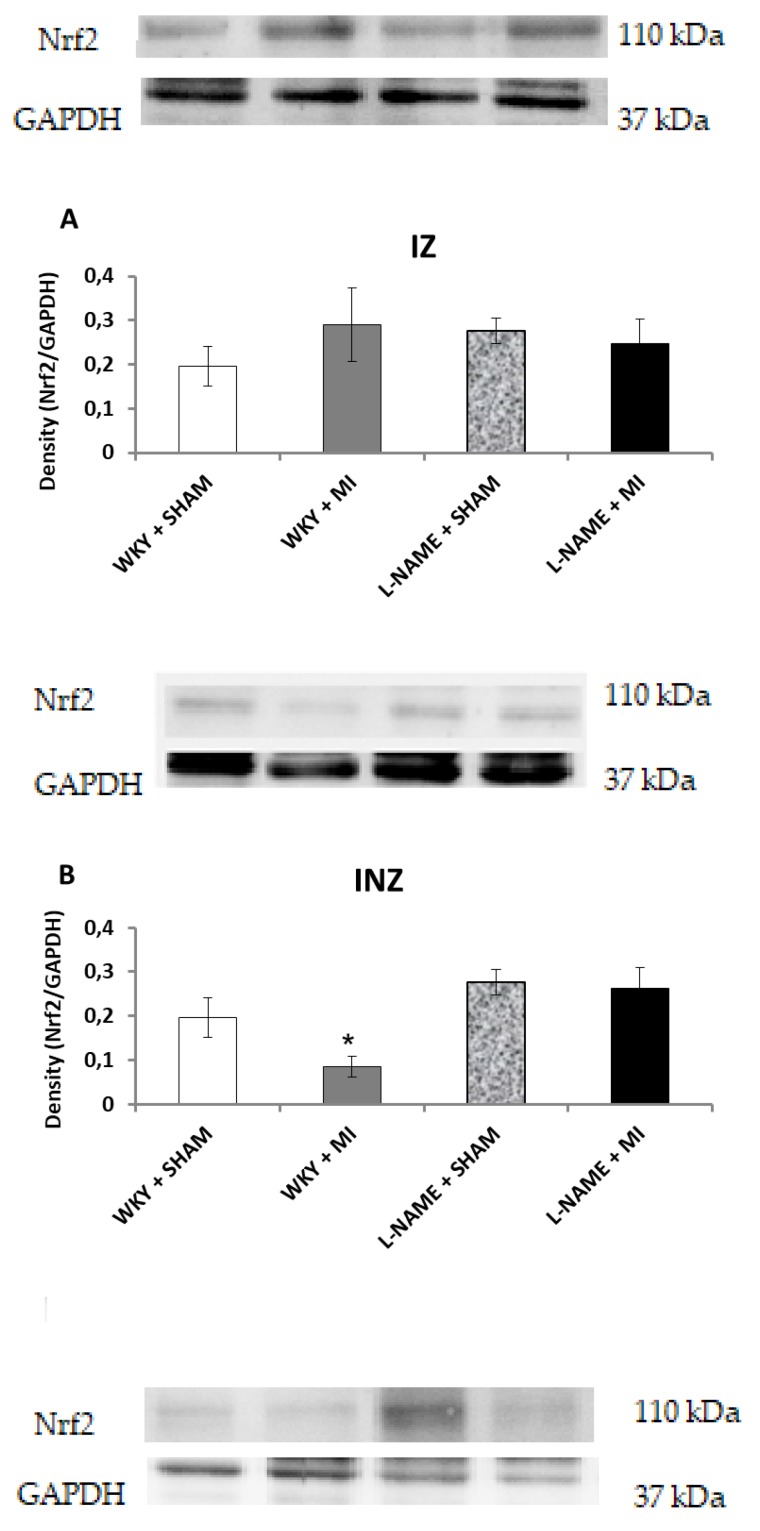
The effect of myocardial infarction on Nrf2 expression of infarcted zone (A), injured zone (B) and non-infarcted zone (C) of myocardium. Representative western blot images and relative Nrf2 expression levels of sham operated Wistar Kyoto rats (WKY + Sham); WKY rats with experimentally induced myocardial infarction (WKY + MI); sham operated WKY rats treated with L-NAME (20 mg/kg/day) (L-NAME + Sham) and L-NAME (20 mg/kg/day) treated WKY rats with experimentally induced myocardial infarction (L-NAME + MI) in infarcted zone (IZ) (A), injured zone (INZ) (B) and non-infarcted zone (NIZ) (C). * *p* < 0.05 compared to WKY + Sham group; N = 6 in each group.

**Figure 5 molecules-24-01682-f005:**
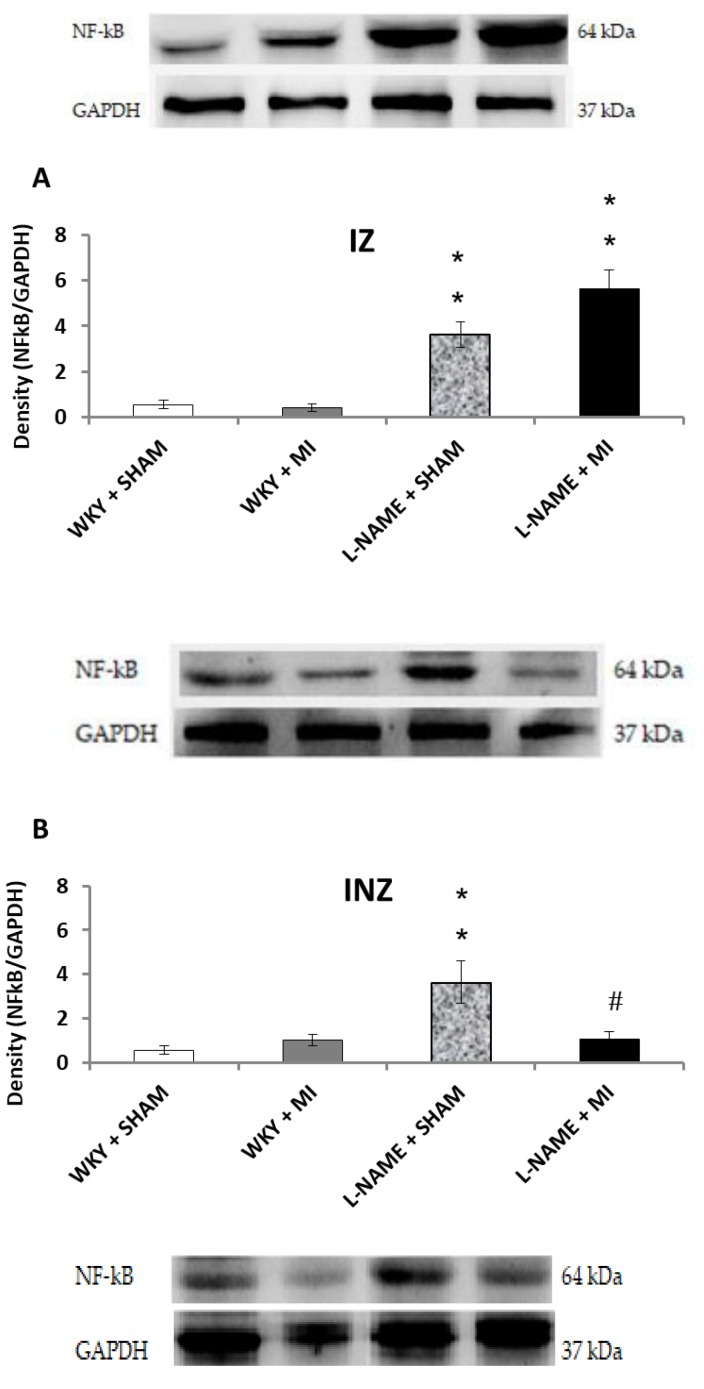
The effect of myocardial infarction on NFҡB/p65 expression of infarcted zone (A), injured zone (B) and non-infarcted zone (C) of myocardium. Representative western blot images and relative NFҡB/p65 expression levels of sham operated Wistar Kyoto rats (WKY + Sham); WKY rats with experimentally induced myocardial infarction (WKY + MI); sham operated WKY rats treated with L-NAME (20 mg/kg/day) (L-NAME + Sham) and L-NAME (20 mg/kg/day) treated WKY rats with experimentally induced myocardial infarction (L-NAME + MI) in infarcted zone (IZ) (A), injured zone (INZ) (B) and non-infarcted zone (NIZ) (C). ** *p* < 0.01 compared to WKY + Sham group; # *p* < 0.05 compared to L-NAME + Sham group; N = 6 in each group.

**Figure 6 molecules-24-01682-f006:**
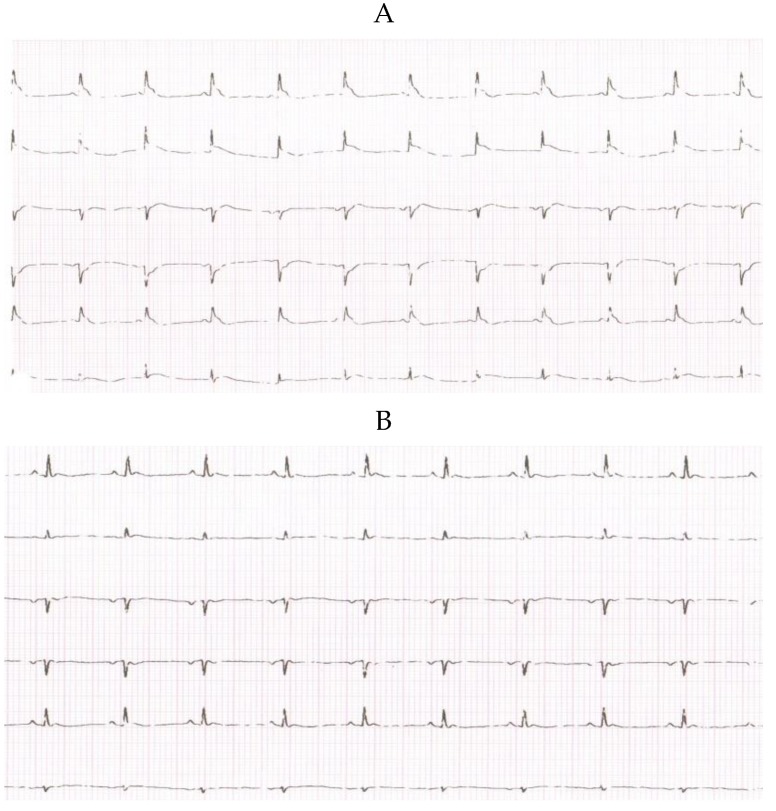
Representative ECG recording. Representative ECG recording of a sham operated Wistar Kyoto rat (6A) and a WKY rat with experimentally induced myocardial infarction (6B).

**Table 1 molecules-24-01682-t001:** Blood pressure of sham operated Wistar Kyoto rats (WKY + Sham); WKY rats with experimentally induced myocardial infarction (WKY + MI); sham operated WKY rats treated with L-NAME (20 mg/kg/day) (L-NAME + Sham) and L-NAME (20 mg/kg/day) treated WKY rats with experimentally induced myocardial infarction (L-NAME + MI). Week 0 - before treatment; Week 4 – four weeks L-NAME administration and before myocardial infarction, Week 5 – seven days after myocardial infarction; ** *p* < 0.01 compared to WKY + Sham; N = 6 in each group.

	Week 0 (mmHg)	Week 4 (mmHg)	Week 5 (mmHg)
WKY + Sham	128 ± 3	119 ± 5	123 ± 4
WKY + MI	128 ± 3	133 ± 8	130 ± 8
L-NAME + Sham	128 ± 3	158 ± 5 **	161 ± 5 **
L-NAME + MI	128 ± 3	160 ± 3**	163 ± 4 **

**Table 2 molecules-24-01682-t002:** Body weight (BW), heart weight (HW) and left kidney weight (LKW) of sham operated Wistar Kyoto rats (WKY + Sham); WKY rats with experimentally induced myocardial infarction (WKY + MI); sham operated WKY rats treated with L-NAME (20 mg/kg/day) (L-NAME + Sham) and L-NAME (20 mg/kg/day) treated WKY rats with experimentally induced myocardial infarction (L-NAME + MI); N = 6 in each group.

	BW (g)	HW (mmHg)	LKW (g)
WKY + Sham	304.20 ± 8.96	1.183 ± 0.24	1.065 ± 0.03
WKY + MI	301.67 ± 7.57	1.097 ± 0.08	1.186 ± 0.05
L-NAME + Sham	297.00 ± 4.94	1.37 ± 0.03	1.211 ± 0.06
L-NAME + MI	295.00 ± 5.87	1.059 ± 0.01	1.132 ± 0.05

**Table 3 molecules-24-01682-t003:** Tumor necrosis factor α (TNF-α) and interleukin 6 (IL-6) levels of sham operated Wistar Kyoto rats (WKY + Sham); WKY rats with experimentally induced myocardial infarction (WKY+MI); sham operated WKY rats treated with L-NAME (20 mg/kg/day) (L-NAME + Sham) and L-NAME (20 mg/kg/day) treated WKY rats with experimentally induced myocardial infarction (L-NAME + MI); ** *p* < 0.01 compared to WKY + Sham; ## *p* < 0.01 compared to L-NAME + Sham group; N = 6 in each group.

	TNF/α [pg/mL]	IL-6 [pg/mL]
WKY + Sham	12.11 ± 2.41	22.41 ± 3.78
WKY + MI	27.72 ± 3.2 **	40.03 ± 2.51 **
L-NAME + Sham	18.19 ± 1.38	36.43 ± 3.08
L-NAME + MI	41.92 ± 2.71 ##	53.17 ± 3.24 ##

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
