# Peer review of "Different adaptive NO-dependent Mechanisms in Normal and Hypertensive Conditions"

_molecules, 2019, doi:10.3390/molecules24091682_

Round 1
Reviewer 1 Report
1. Week “0” data of BP of all groups are similar. It should be checked.
2. 160,63 ± 5.48 should be 160.63 ± 5.48
3. 163,13 ± 4.15 should be 163.13 ± 4.15
4. L-NAME Sham should be L-NAME + Sham across a manuscript
5. The basis of the dose of L-NAME should be provided.
6. L-NAME (Sigma-Aldrich) was 271 dissolved in the drinking water in the dose of 20 mg/kg/ day and administered daily to 272 12-week-old normotensive rats for 4 weeks. How do the authors ensure that all the rats drink water? Is there any difference in water consumption pattern?
7. The purpose of administering Butorphanol in the dose 2 mg/kg s.c.+ 2 mg/kg meloxicam should be indicated.
8. Any specific reason for Duncan´s test over other posthoc analysis.
9. 12-week-old Wistar Kyoto rats or 16-week old rats, Avoid any ambiguity.
10. Any specific purpose of measuring kidney weight. Do the biochemical parameters were analyzed in the kidney or kidney function test.
11. An overdose of anaesthesia, which anaesthesia, route and dose should be specified.
12. The duration of ligation and duration of reperfusion should be provided.
13. THE ECG recordings can be provided. It will be vital to providing an important marker of myocardial injury.
14. Did the authors measure CK_MB, LDH or troponin, cTnI, cTnT or any specific biomarker? Either ECG or one biochemical biomarker should be included in order to provide evidence of myocardial injury.
15. Do the authors performed a histopathological study on the heart.
16. Do the authors performed a TTC staining of the heart.
17. Do the authors have ventricular data?
18. Some data either on ECG, biomarkers or histology will add evidence-based value to the study and its outcome.
19. It's better to represent BP in single digits like 126 or 157, avoid decimal points.
20. Figure 5, Panel C thewestern cropping is not appropriate. Replace the figure.
21. It is not very novel but not ignorant and can add knowledge to the present known. The authors should improve the manuscript for acceptance.
Author Response
We would like to thanks reviewer for reading our manuscript. We highly appreciate the reviewer`s interest in our research, favorable comments and constructive suggestions. Our manuscript was checked by a native English speaker. We corrected and added all suggested points. All changes and additions (except English corrections) in revised manuscript are highlighted in yellow for ease of review.
Comments and Suggestions for Authors
1. Week “0” data of BP of all groups are similar. It should be checked.
Response: In week 0 all animals were measured as a one group. Thereafter, animals were randomly devided into four investigated groups. Therefore, we have the same value of blood pressure in each group of this week.
2. 160,63 ± 5.48 should be 160.63 ± 5.48
3. 163,13 ± 4.15 should be 163.13 ± 4.15
Response: In point 19, the reviewer also recomend using single digits, we corrected it in the table 1 and instead of decimal digits we used single digits.
4. L-NAME Sham should be L-NAME + Sham across a manuscript
Response: We corrected L-NAME + Sham across the manuscript as well as in figures.
5. The basis of the dose of L-NAME should be provided.
Response: The dose of L-NAME was the same during experiment – 20 mg/kg/day dissolved in drinking water.
6. L-NAME (Sigma-Aldrich) was dissolved in the drinking water in the dose of 20 mg/kg/ day and administered daily to 12-week-old normotensive rats for 4 weeks. How do the authors ensure that all the rats drink water? Is there any difference in water consumption pattern?
Response: Prior to experiment, we measured the water consumtion. At the begining, they drunk 30ml water per day, aproximately. We dissolved specific amount of L-NAME in this volume. We adjusted the volume of water if it was neccesary. There was no difference between any groups in water consumtion.
7. The purpose of administering Butorphanol in the dose 2 mg/kg s.c.+ 2 mg/kg meloxicam should be indicated.
Response: The purpose of using Butorphanol and Meloxicam was postoperative analgesia. Multimodal or balanced analgesia involves to use a combination of analgesic drugs from different classes to yield additive analgesic effects. The combination of analgesic drugs can decrease the side effects of each drug and at the same time increase the efficacy. In this case there is a synergistic analgetic effect of opioid (butorphanol) and antiinflammatory drug (meloxicam).
8. Any specific reason for Duncan´s test over other posthoc analysis.
Response: Duncan test is suitable for statistical calculation of our results. It involves the computation of numerical boundaries, that allow for the classification of the difference between any 2 treatment means or conditions as significant or non-significant. This requires calculation of a series of values each corresponding to a specific set of pair comparisons. That is the reason why we are using also Duncan`s test.
9. 12-week-old Wistar Kyoto rats or 16-week old rats, Avoid any ambiguity.
Response: We corrected and specified exactly that we used 16-week-old rats for experimentaly induced MI with L-NAME induced hypertension from the week 12th of the age.
10. Any specific purpose of measuring kidney weight. Do the biochemical parameters were analyzed in the kidney or kidney function test.
Response: Since hypertension can be related to kidney failures, we checked the kidney weight as well as basic morphology of kidney. Unfortunately, we did not measure any functional parameters.
11. An overdose of anaesthesia, which anaesthesia, route and dose should be specified.
Response:: We added to the manuscript which anaesthesia, route and dose were used to euthanase the animals (i.p. with titelamine-zolazepan 60mg/kg).
12. The duration of ligation and duration of reperfusion should be provided.
Response:: The duration of coronary artery ligation lasted 20 minutes. Reperfusion lasted 7 days.
13. THE ECG recordings can be provided. It will be vital to providing an important marker of myocardial injury.
Response: We added representative ECG records to manuscript to provide an evidence of myocardial infarction. It is shown as a Figure 6 in methodology part.
14. Did the authors measure CK_MB, LDH or troponin, cTnI, cTnT or any specific biomarker? Either ECG or one biochemical biomarker should be included in order to provide evidence of myocardial injury.
Response: We added representative ECG records to manuscript. Unfortunately, we did not see any differences in troponin I and creatinin kinase levels between experimntal groups 7 days after MI. According the guidance, increased troponin I level occurs approximately 3-8 hours after IM, with reaching a maximum after 10-20 hours and lasted approximately 15-120 hours after IM. CK values increase about 3-9 hours after MI with peaks at 10-20 hours after MI, and return to normal after about 72 hours (Danese and Montagnana, 2016). We did not want to anesthetize the animals again to take a blood 20 hours after surgery. 7 days could be too long to see any differences in the troponin level.
15. Do the authors performed a histopathological study on the heart.
Response: Regrettably, we did not perform any histopathological study. We oriented on biochemical parameters after MI.
16. Do the authors performed a TTC staining of the heart.
Response: Unfortunately, we were not able to do TTC staining together with other biochemical measurements.
17. Do the authors have ventricular data?
Response:: Unfortunatelly, we did not measure any ventrical data.
18. Some data either on ECG, biomarkers or histology will add evidence-based value to the study and its outcome.
Response: We added representative ECG records to manuscript to provide an evidence of myocardial infarction. It is shown as a Figure 6 in methodology part.
19. It's better to represent BP in single digits like 126 or 157, avoid decimal points.
Response: According reviewer`s recomendation we corrected the table 1 and instead of decimal digits we used single digits.
20. Figure 5, Panel C the western cropping is not appropriate. Replace the figure.
Response:The figure 5C was replaced.
21. It is not very novel but not ignorant and can add knowledge to the present known. The authors should improve the manuscript for acceptance.
Response: Our manuscript was checked by a native English speaker. We also corrected and added all suggested points from reviewers to imrove the manuscript.
Reviewer 2 Report
Well written manuscript, still needs to do some proofreading.
More proof readings needs to be done.
Did authors checked cardiac troponin I levels in treated mice?
Author Response
We would like to thanks reviewer for reading our manuscript. We highly appreciate the reviewer`s interest in our research, favorable comments and constructive suggestions. Our manuscript was checked by a native English speaker. All changes and additions in revised manuscript are highlighted in yellow for ease of review.
Comments and Suggestions for Authors
Well written manuscript, still needs to do some proofreading.
More proof readings needs to be done.
Response: We corrected English of the manuscript. We also added representative ECG records to manuscript to provide an evidence of myocardial infarction. It is shown as a Figure 6 in methodology part.
Did authors checked cardiac troponin I levels in treated mice?
Response: Unfortunately, we did not see any differences in troponinI levels between experimntal groups 7 days after MI. According the guidance, increased troponin I level occurs approximately 3-8 hours after IM, with reaching a maximum after 10-20 hours and lasted approximately 15-120 hours after IM (Danse and Montagnana, 2016). We did not want to anesthetize the animals again to take a blood 20 hours after surgery. 7 days could be too long to see any differences in the troponin level.

Reviewer 3 Report
In the study titled, "Different adaptive NO-dependent mechanisms in normal and hypertensive conditions", Kosutova M et al., describe the effect of NO blockade on myocardial infarction injury in rats. Overall, hypothesis of the paper is unclear and there is no mechanism for the claims. L-NAME has been widely used for NO-blockade with adverse effects on myocardial structure and function. The rationale for blocking NO 4 weeks prior remains unclear.
Author Response
We would like to thanks reviewer for reading our manuscript. We highly appreciate the reviewer`s interest in our research, favorable comments and constructive suggestions. Our manuscript was checked by a native English speaker. All changes and additions in revised manuscript are highlighted in yellow for ease of review.
Comments and Suggestions for Authors
In the study titled, "Different adaptive NO-dependent mechanisms in normal and hypertensive conditions", Kosutova M et al., describe the effect of NO blockade on myocardial infarction injury in rats. Overall, hypothesis of the paper is unclear and there is no mechanism for the claims. L-NAME has been widely used for NO-blockade with adverse effects on myocardial structure and function. The rationale for blocking NO 4 weeks prior remains unclear.
Response: As stated in the introduction, it is generally accepted, that NO is an essential signaling molecule involved in many physiological processes in animals and humans, including neurotransmission, hypertension and heart failure. It is also a known trigger and mediator for cardioprotection [14], reducing myocardial necrosis and apoptosis [15, 16]. On the other hand, in pathophysiological conditions such ischemia, the accumulation of NO from both enzymic and non-enzymic sources may play a significant role in I/R injury by increasing production of reactive species [17]. Therefore, it is of interest to use a model of L-NAME induced hypertension for experimental myocardial infarction. We expected that, on the one hand, the unfavorable conditions of hypertension due to 4 weeks of NO synthase inhibition significantly worsen the consequences of myocardial infarction. On the other hand, lack of nitric oxide could paradoxically have a positive effect on heart ischemia. Of course, we expected, that upregulation of different NOS isoforms would play a significant role in this process.
Indeed, our results indicated that eNOS upregulation in INZ after MI in L-NAME rats may contribute to increased NOS activity and serve as a compensatory mechanism improving perfusion of the myocardium. Conversely, iNOS expression was increased in IZ of these animals and contributed to increased NOS activity in this zone which may lead to inflammatory process and necrotic changes.
Round 2
Reviewer 3 Report
Authors response is duly noted, however, the manuscript lack a central hypothesis connecting NO inhibition with the signaling mechanism proposed.
Author Response
Authors response is duly noted, however, the manuscript lack a central hypothesis connecting NO inhibition with the signaling mechanism proposed.
Response: We would like to thanks reviewer for reading our manuscript and response. We added our hypothesis into introduction and highlighted in yellow.
Chronic inhibition of NOS by NG-nitro-L-arginine methyl ester (L-NAME) is a well-established model of experimental hypertension and organ damage characterized by myocardial hypertrophy, fibrosis and coronary artery wall hyperplasia [20-23]. Myocardial infarction prognosis is conditioned by a series of risk factors which rend the process more incisive. If hypertension present at time of myocardial infarction, worsens the prognosis of MI by inducing a burden of oxidative and inflammatory mediators released within the heart. Therefore, studies with molecules possessing multiple activities and thus capable of blocking or reversing the pathological progress of hypertension and MI at different levels are needed. Nitric oxide represents an important part of this strategy. Generally, NO is an essential signaling molecule involved in many physiological and pathophysiological processes. Therefore, it is of interest to use a model of L-NAME induced hypertension for experimental myocardial infarction. On the one hand, the unfavorable conditions of hypertension due to 4 weeks of NO synthase inhibition may significantly worsen the consequences of myocardial infarction. On the other hand, lack of nitric oxide could paradoxically have a positive effect on heart ischemia. Therefore, the aim of the study was to block NO production with L-NAME administration 4 weeks prior to experimentally induced myocardial infarction by coronary artery ligation and thereafter to evaluate the effect of NO deficiency on selective biochemical parameters within individual myocardial zones.